# Pan-Genomic Sequencing Reveals Actionable *CDKN2A/2B* Deletions and Kataegis in Anaplastic Thyroid Carcinoma

**DOI:** 10.3390/cancers13246340

**Published:** 2021-12-17

**Authors:** Adam Stenman, Minjun Yang, Johan O. Paulsson, Jan Zedenius, Kajsa Paulsson, C. Christofer Juhlin

**Affiliations:** 1Department of Molecular Medicine and Surgery, Karolinska Institutet, 17176 Stockholm, Sweden; adam.stenman@ki.se (A.S.); jan.zedenius@ki.se (J.Z.); 2Department of Breast, Endocrine Tumors and Sarcoma, Karolinska University Hospital, 17176 Stockholm, Sweden; 3Department of Laboratory Medicine, Division of Clinical Genetics, Lund University, 22185 Lund, Sweden; minjun.yang@med.lu.se (M.Y.); kajsa.paulsson@med.lu.se (K.P.); 4Department of Oncology-Pathology, Karolinska Institutet, 17176 Stockholm, Sweden; johan.paulsson@ki.se; 5Department of Pathology and Cancer Diagnostics, Karolinska University Hospital, 17176 Stockholm, Sweden

**Keywords:** anaplastic thyroid carcinoma, whole-genome sequencing, CDKN2A, CDKN2B, kataegis, molecular targets

## Abstract

**Simple Summary:**

Anaplastic thyroid carcinoma (ATC) is a tumor with exceedingly high mortality rates, and current treatment options are very limited once the disease has spread beyond the thyroid gland. Most of what we know of ATCs from a genetic standpoint is based upon previous analyses in which limited DNA segments are sequenced, and little is known about the so-called non-coding DNA regions. To identify novel genetic mechanisms of potential therapeutic use, we sequenced the entire genome of eight ATC cases and corresponding normal tissues. We found a recurrent deletion of two neighboring genes responsible for inhibiting cell division, and these findings may be of clinical interest as there are specific drugs available for tumors with this gene aberration. We also found regional aggregations of mutations in specific clusters, a phenomenon entitled “kataegis”, which in turn could be therapeutically relevant.

**Abstract:**

Anaplastic thyroid carcinoma (ATC) is a lethal malignancy characterized by poor response to conventional therapies. Whole-genome sequencing (WGS) analyses of this tumor type are limited, and we therefore interrogated eight ATCs using WGS and RNA sequencing. Five out of eight cases (63%) displayed *cyclin-dependent kinase inhibitor 2A* (*CDKN2A*) abnormalities, either copy number loss (n = 4) or truncating mutations (n = 1). All four cases with loss of the *CDKN2A* locus (encoding p16 and p14arf) also exhibited loss of the neighboring *CDKN2B* gene (encoding p15ink4b), and displayed reduced *CDKN2A/2B* mRNA levels. Mutations in established ATC-related genes were observed, including *TP53*, *BRAF*, *ARID1A*, and *RB1*, and overrepresentation of mutations were also noted in 13 additional cancer genes. One of the more predominant mutational signatures was intimately coupled to the activity of Apolipoprotein B mRNA-editing enzyme, the catalytic polypeptide-like (APOBEC) family of cytidine deaminases implied in kataegis, a focal hypermutation phenotype, which was observed in 4/8 (50%) cases. We corroborate the roles of *CDKN2A/2B* in ATC development and identify kataegis as a recurrent phenomenon. Our findings pinpoint clinically relevant alterations, which may indicate response to CDK inhibitors, and focal hypermutational phenotypes that may be coupled to improved responses using immune checkpoint inhibitors.

## 1. Introduction

Anaplastic thyroid carcinoma (ATC) is a highly lethal undifferentiated malignant lesion that often affects older patients [1]. The tumor generally develops through dedifferentiation from a pre-existing well-differentiated thyroid carcinoma (papillary or follicular thyroid carcinoma; PTC/FTC), a hypothesis that has been reinforced by modern genetic analyses [2,3,4,5,6]. The clinical presentation is often dramatic due to the invasive-prone properties of the tumor, with rapid onset of symptoms that may include a pronounced swelling of the neck, sudden hoarseness, and dyspnea. The treatment is mostly of palliative nature, and debulking surgery as well as adjuvant regimes of combined radio- and chemotherapy are usually needed to reduce the tumor burden and reduce the risk of an acute airway obstruction [7,8]. Although subsets of patients with small tumors limited to the thyroid gland can be cured following the timely excision of all primary tumor tissue, metastatic dissemination is common and almost always associated to poor outcomes [7]. In all, very few patients survive if the ATC spreads to distant sites, although extraordinary outcomes have been achieved in single cases [9,10]. The underlying genetic mechanisms of ATCs have been partly elucidated through comprehensive pan-exomic landscape studies [11,12,13]. Since most ATCs arise as a consequence of accumulating genetic aberrations in a pre-existing well-differentiated thyroid carcinoma, many ATCs harbor mutations usually associated with the development of PTCs and FTCs, such as those targeting the genes *BRAF*, *NRAS*, *HRAS*, *KRAS*, *PIK3CA*, *E1F1AX*, and *PTEN* [2,11,12,13]. Additionally, *TERT* promoter mutations, a general predictor of worse clinical outcomes in thyroid tumors, are highly prevalent in ATCs. Moreover, subsets of cases display a hypermutation phenotype, which is coupled to somatic mutations in mismatch repair genes (MMRs) [11,12]. Regarding chromosomal alterations, copy number losses and mutations of *CDKN2A* and *CDKN2B* as well as amplifications of *CCNE1*, *KDR*, *KIT*, and *PDGFRA* have been described [12]. In terms of epigenetic events, the ATC genome is generally hypomethylated, with focal hypermethylation of several thyroid genes coupled to thyroid differentiation [14].

In one of the most comprehensive genetic analyses of ATCs to date (n = 196), Pozdeyev and colleagues used a focused gene sequencing panel and triaged the tumors into four main mutational clusters (clusters 1–4) [15]. Cluster 1 includes tumors with *BRAF* V600E mutations occurring with or without combinations of mutations in *PIK3CA*, *AKT1*, or *ARID2*. Given the intimate association between *BRAF* mutations and PTC development, ATCs in cluster 1 are believed to stem from pre-existing PTCs. Cluster 2 ATCs are characterized by loss-of-function alterations in *CDKN2A* and *CDKN2B*, in addition to additional mutations normally associated with either PTC or FTC development. Cluster 3 is amassed in NRAS mutations and therefore believed to contain ATCs developing mainly from FTCs. Finally, cluster 4 is highly heterogeneous, containing ATCs with hypermutation profiles and MMR gene mutations, as well as cases with *PTEN*, *NF1*, and *RB1* mutations, as well as amplifications of *KDR*, *KIT*, and *PDGFRA* [15].

Although the above-mentioned advances have vastly increased our understanding of ATCs, the bulk of genetic data derived from ATCs is based on focused gene panels or whole-exome sequencing (WES) analyses. To build on this, we performed whole-genome sequencing (WGS) and global RNA sequencing using an ATC cohort with established clinical data to identify non-coding mutations, mutational signatures, and chromosomal alterations not readily distinguishable without the use of pan-genomic sequencing.

## 2. Materials and Methods

### 2.1. Patient Material

All patients had been previously diagnosed and operated on at the Karolinska University Hospital during 1989–2007. All patients were diagnosed by fine needle aspiration biopsies and subsequently treated with neoadjuvant combined radio- and chemotherapy before debulking surgery was commenced. One of the patients (105) survived and was free of relapse and metastases at follow up more than 200 months after surgery and died of another cause 20 years later, while the other patients died of disease between 2 weeks and 4 months after surgery. Tumor sizes varied between 4 and 11 cm with 7.5 cm on average. None of the patients presented with distant metastases at the time of surgery. Histopathological assessments were performed according to the criteria laid down by the most recent World Health Organization (WHO) criteria at the time of the diagnosis. Brief histopathological and clinical parameters of the cohort are detailed in Table 1.

### 2.2. DNA Extraction and Tissue Representativity Testing

Fresh frozen tumor tissue from primary tumors was collected from our biobank repositories, and extraction of DNA and RNA was performed using the DNeasy Blood & Tissue Kit and RNeasy Plus Mini Kit, respectively (both Qiagen, Hilden, Germany). Moreover, a piece of the same tissue part was saved for formalin fixation followed by embedment in paraffin and sectioning for tumor tissue content analysis using light microscopy. The tumor cell percentages were on average 70%, ranging from 50% to 90%. As normal controls, leukocyte DNA was ordered from the blood biobank at Karolinska Institutet for 1 patient (110). This blood was preoperatively drawn from the patient and saved in a −86 °C freezer. If no peripheral blood was available, DNA was extracted from fresh frozen normal thyroid tissue (muscular tissue for case 109) acquired in conjunction with the dissection of the primary tumor (n = 7). These normal tissue samples were also investigated regarding representation testing and contained more than 80% normal cells and were free of cancer cells.

### 2.3. Whole-Genome Sequencing (WGS)

WGS was performed at the National Genomics Infrastructure Sweden (SciLifeLab, Stockholm, Sweden). In short, library preparation with one microgram of DNA was performed using the Illumina TruSeq PCR-Free workflow with a target insert size of 350 base pairs. Samples were sequenced using the NovaSeq 6000 S4 platform.

### 2.4. RNA Sequencing

Library preparation was performed using the Illumina TruSeq strand-specific RNA libraries using poly-A selection from a total of X RNA samples, and subsequently multiplex sequenced on the Illumina NovaSeq S4, followed by demultiplexing and gold standard quality control.

### 2.5. WGS Data Analysis

Paired-end sequencing (2 × 150) was done to ~60× coverage for diagnostic samples and ~30× coverage for remission. Sequencing reads were aligned to the reference genome human_g1k_v37 using the Burrows-Wheeler Aligner (version 0.7.17) [16]. Somatic variants were identified by using the GDC DNA-Seq analysis pipeline (https://docs.gdc.cancer.gov/Data/Introduction/, accessed on 12 November 2021). Mutations that passed the internal filters of the variation caller were further filtered by a minimum depth of 20 reads and a minimum variant read depth of 5 reads. The variant files (vcf file) were converted into maf files by the vcf2maf package (https://github.com/mskcc/vcf2maf, accessed on 12 November 2021) and annotated by the Ensembl Variant Effect Predictor [17]. Significantly mutated genes were identified by MutSigCV (version 1.41) with *p* < 0.05 by using all informative mutations as input [18]. Somatic structural variants (SVs) were identified by Manta [19], DELLY [20], SvABA [21], and GRIDSS [22] with default settings, then merged using MAVIS [23]. Patchwork was applied to ascertain somatic copy number abnormalities (CNAs) [24]. For all software used in the bioinformatics work-flow of both DNA and RNA analyses, analyses and database access (whenever applicable) were performed between April 2020 and May 2021.

### 2.6. RNA Sequencing Data analysis

For expression analysis, RNA sequencing reads were aligned to the reference genome human_g1k_v37 using STAR, read counts for each gene were obtained by using GDC mRNA analysis pipeline (https://docs.gdc.cancer.gov/Data/Introduction/, accessed on 12 November 2021), and gene expression values were estimated by RSEM [25]. The GATK ASEReadCounter was used to identify the expressed SNVs that were found by WGS [26]. To identify fusion transcripts, FusionCatcher (https://www.biorxiv.org/content/10.1101/011650v1, accessed on 12 November 2021), InFusion [27], and Arriba [28], were used. The list of fusion genes was filtered to remove chimeras that were identified as read-through transcripts, pseudogenes, unannotated genes, and fusions between gene family members.

### 2.7. Mutational Signatures and Kataegis

The R package MutationalPatterns [29] was used to decompose mutational profiles into pre-defined single base substitution (SBS) mutational signatures based on the Sanger mutational signatures (v3.1—June 2020, https://cancer.sanger.ac.uk/signatures/, accessed on 12 November 2021) and to ascertain the relative contributions of the SBS mutational signatures in each case. Regions with kataegis, a recently described phenomenon of localized hypermutation in cancer, were identified as those segments containing six or more consecutive mutations within a mean inter-mutation distance of 1000 bp by using Maftools [30,31].

### 2.8. Fusion Gene Validation

Primers flanking the fusion break were designed using Primer3, with primer sequences F: 5′-GTGGGTACAGATCAGAAGAGC and R: 5′-CATGAATTGGCCAGTGGACA. Genomic DNA from the fusion gene sample, an additional case from the cohort, and two unrelated normal thyroid samples were used for amplification with a Platinum II hot start PCR kit (Thermo-Fisher Scientific, Waltham, MA, USA). The fusion gene product was validated with gel electrophoresis and Sanger sequencing.

### 2.9. Gene Ontology Analyses

In order to identify common denominators in terms of signaling networks when analyzing genes with SNVs in promoter or enhancer regions, the Reactome Pathway Browser was used (https://reactome.org/, accessed on 12 November 2021).

## 3. Results

### 3.1. Whole-Genome Sequencing Quality Parameters

In total, 16 samples were successfully sequenced using WGS and the Illumina HighSeq X technology, including 8 primary ATCs and 8 corresponding constitutional tissues. The total million reads per tumor and normal sample was 817 and 423 on average, respectively. The aggregated percentage of bases that had a quality score more than the Q30 value was 90% on average, ranging from 87.06–92.19%. The Q30 mark is equal to an inferred base call accuracy of 99.9%.

### 3.2. Somatic Mutational Overview

Using the Illumina HiSeq X platform, a total of 41,837 (91.4%) single nucleotide variants (SNVs) and 3960 (8.6%) small insertions/deletions (indels) were found in the eight investigated cases, corresponding to 0.2–6.3 SNVs/indels per Mb/case and with a median of 3031 SNVs/indels per case (range 596–18,912) (Appendix A). The numbers of non-silent SNVs/indels in coding regions were 319 and 26, respectively, for all 8 cases. Transition-type SNVs (especially C > T transitions) were overrepresented in our ATC cohort (Figure 1). Of the SNVs, 588 (1.4%) were also detected at the transcriptome level, with a median of 27 SNVs per case (range 1–329) in matched RNA sequencing data, of which 151 were non-silent SNVs (Appendix A). Known oncogenic mutations that were also found in the RNA sequencing data included, for example, a *BRAF* c.1799T>A (p.V600E) mutation in case 102T, a *BRAF* c.1406G>C (p.G469A) mutation in case 105T, and an *NRAS* c.182A>G (p.Q61R) mutation in case 101T. Missense mutations in several tumor suppressor genes were also detected, for instance, *TP53* mutations in cases 101T and 108T, and *PTEN* and *NF2* mutations in case 110T (Figure 1).

### 3.3. Aberrations in Genes Commonly Mutated in Anaplastic Thyroid Cancer

To highlight mutational events in established genes, we compared our complete list of somatic mutations with the top 20 genes mutated in ATCs as reported by the Catalogue of Somatic Mutations in Cancer (COSMIC) database (Figure 1). All ATCs except two cases (103 and 109) exhibited one or several mutations in established thyroid-related genes, with *TP53* (three cases), *TERT* promoter (three cases with established C228T or C250T mutations), and *BRAF* (two cases) as the top three mutated thyroid-related genes in our cohort.

### 3.4. CDKN2A/B

In our cohort, we found a single ATC with a deleterious *CDKN2A* mutation as well as four additional tumors with gene deletions, and *CDKN2A* therefore constitutes the most aberrantly affected cancer-related gene in the entire cohort. Of note, all four ATCs with *CDKN2A* gene deletions also exhibited concomitant *CDKN2B* gene deletions. When comparing these findings to the RNA sequencing output, we conclude that *CDKN2A*/*B* mRNA levels were significantly (*p* < 0.05, Mann–Whitney U) downregulated as compared to diploid cases (Figure 2).

### 3.5. Novel Genes of Interest

By MutSig analyses, we found a significant overrepresentation of non-silent somatic mutations in 14 cancer-related genes, namely *TP53, THY1, CYBA, THAP1, MYOZ3, RPS5, IL9R, CNPY4, FCGR3A, RAB40A, RNASEH1, CDKN2A, ACAT1*, and *CLEC11A*. While *TP53* and *CDKN2A* exhibit known roles in ATC development, many of these genes are unknown from a thyroid perspective [11,15].

### 3.6. Mutational Signatures and Kataegis

Using mutational profiling, we found that SBS2 and SBS13 (https://cancer.sanger.ac.uk/cosmic/signatures/SBS/SBS13.tt, accessed on 12 November 2021) were two of the top five mutational signatures in our cohort (Figure 1 and Figure 3). These signatures are intimately coupled to the activity of Apolipoprotein B mRNA-editing enzyme, catalytic polypeptide-like (APOBEC) family of cytidine deaminases implied in kataegis, a local hypermutation phenotype, which was observed in 4/8 cases. We identified 14 hypermutated genomic regions in these four cases (101T, 103T, 105T, and 110T) and a high fraction of C > T mutations (59/106) was observed in kataegis regions for all informative cases (Figure 4, Appendix A). Moreover, 4 of the top 10 mutational signatures were associated with DNA repair (SBS3, SBS30, SBS9, and SBS6), thus reinforcing the established connotation between defective DNA repair mechanisms, hypermutability, and the development of ATCs [5,6,11,32].

### 3.7. Non-Coding Mutations in Introns, Promoters, and Enhancers

One of the benefits of employing WGS as opposed to exome-capture based methodology is the ability to screen for SNVs occurring in non-coding elements with potential for tumor development. By extensive screening of promoter or transcription binding sites, we found 960 SNVs directly located within these regions across the 8 ATCs. Of these, only two SNVs were recurrent (two established C250T mutations in the *TERT* promoter). Conventional *TERT* promoter mutations were found in five cases, including established mutations in three tumors (chr 5:1295228 C > T; commonly denoted “C228T” in a single case, chr 5:1295250 C > T; “C250T” in two additional cases). Moreover, one additional case each harbored a chr 5:1295105 C > T (“C105T”, case T105) and a chr 5:1295168 C > T (“C168T”, case T101) alteration, respectively. These two unconventional *TERT* promoter SNVs were mutually exclusive with the C228T and C250T mutations and were not annotated in the database of SNP (dbSNP), thereby possibly affecting variants with tumorigenic potential. While C168T is located within the *TERT* promoter core region (also harboring the C228 and C250 regions), C105T is located just upstream of the ATG start site of the *TERT* gene.

Using the Reactome Pathway Browser to analyze the gene pool with SNVs in promoter or enhancer regions, a significant enrichment of genes associated with Insulin-like Growth Factor-2 (IGF-2) mRNA-Binding Protein (IGF2BPs/IMPs/VICKZs) pathways, RUNX3 regulation of NOTCH signaling, and presynaptic depolarization and calcium channel opening signaling were found. Mutated gene promoters associated with the IGF2 signaling pathway included *ACTB*, *IGF2*, and *CD44*.

### 3.8. Structural Variants

To analyze structural variants, we selected high-quality fusion events reported by Manta, DELLY, GRIDSS, and SvABA. In total, we identified 399 structural events, of which 21 were denoted as high-confidence events also reproduced by RNA sequencing (Appendix A). When manually scrutinizing these 21 high-confidence events, we found a novel *Exportin 5 (XPO5)*—*Carbohydrate Sulfotransferase 9 (CHST9)* fusion in case 101T. This fusion caught our attention as *XPO5* is identified as an oncoprotein in several human cancers [33,34]. The fusion was verified using direct PCR and Sanger sequencing (Figure 5).

### 3.9. Potential Therapeutic Targets

From a clinical standpoint in terms of ATC genetics, targetable mutations for precision medicine purposes would be the most urgent matter. We therefore screened the mutational output for potentially druggable targets using the Drug Gene Interaction Database (https://dgidb.org/, accessed on 12 November 2021). The results from this analysis are available as Appendix A. We found 1990 compounds with the potential to interact with 84 mutated genes distributed among all ATCs in this cohort, including, among others, *APOB, ARID1A, BRAF, CDKN2A, KIT, PIK3CA*, and *PTEN*.

### 3.10. Transcriptome Sequencing

The aggregated percentage of bases that had a quality score more than the Q30 value was 91.8%. Expression levels could be ascertained for 19,011 mRNAs in 9 ATC cases. The output was then used to filter for expressed structural variants (Appendix A).

## 4. Discussion

Next-generation sequencing of ATCs has greatly increased our general understanding of this entity, allowing identification of common molecular driver events, a shared clonal ancestry with well-differentiated thyroid cancer, as well as several target options for individualized treatments. As the bulk of data stems from focused NGS panels or WES analyses, we sought to expand the field by a comprehensive WGS screening. We corroborate previous findings of recurrent *CDKN2A/CDKN2B* gene deletions, in turn coupled to downregulation of the corresponding gene products. We also reveal focal genomic regions containing hypermutated sequences, a phenomenon that could possibly be linked to the observed overrepresentation of mutational signatures involving cytidine deaminases. Moreover, a novel fusion involving the *XPO5* oncogene is described in a single ATC, as well as two cases with equivocal *TERT* promoter mutations not previously reported in ATCs. Potentially druggable targets were found in all cases, thus reinforcing the need for sequencing analyses when diagnosing ATCs in the clinical setting.

Six out of eight cases (75%) exhibited at least one mutation in established thyroid-related genes, including, among others, *TP53*, the *TERT* promoter, *BRAF, ARID1A, FGFR1, NRAS*, and *PIK3CA*. Moreover, MutSig analyses revealed a significant overrepresentation of somatic mutations in 14 cancer-related genes, including several genes without an established role in ATC development. Of these genes, *Ribonuclease H1 (RNASEH1)* caught our attention given its established role in DNA replication and repair [35] and the recognized role of defective DNA repair in thyroid cancer progression [5,6]. In our cohort, case 105 exhibited an *RNASEH1* nonsense mutation with damaging properties, but given the other known driver gene alterations in this sample (*CDKN2A/B* deletions and a *BRAF* mutation), a passenger status cannot be excluded. Intriguingly, two ATC cases (T103 and T109) were genetic orphans as they did not exhibit mutations in either thyroid-related genes or in genes detected through the MutSig analyses. These cases also exhibited fewer structural variants than the rest of the cohort (Appendix A), and no obvious cancer-related fusion gene event was noted. Thus, the genetic etiology of these two cases remains to be established.

ATCs are genetically unstable tumors, and aberrant copy number landscapes as well as gross chromosomal aberrations have been previously characterized [11,36]. Intriguingly, while driver gene fusions (*RET* fusions or *NTRK* fusions in PTCs, *PAX8-PPARγ* fusions in FTCs) are often mutually exclusive with driver gene mutations in well-differentiated thyroid cancer, they are generally absent in ATCs [12,13]. Even so, chromosomal rearrangements in ATCs are abundant [11,36]. We found several structural rearrangements that were correctly identified by a combination of bioinformatics-related software, but only a fraction was also detected in the RNA sequencing data, suggesting that many of these alterations might be “silent” from a transcriptional point of view. Of potential value for further studies, we highlighted the *XPO5-CHST9* fusion, which was observed both on the DNA and RNA level, and was also reproduced using PCR and Sanger sequencing. Of note, the *XPO5* gene encodes exportin 5, a protein needed to transport small RNA molecules from the nuclear compartment to the cytosol, including micro-RNA (miRNA) precursors. Moreover, the *XPO5* protein might also exhibit miRNA-processing functions, enhancing the function of the DROSHA/DGCR8 microprocessor unit [37]. Interestingly, *XPO5* is an established oncogene in colon cancer [34], but whether the *XPO5-CHST9* fusion would cause a similar effect in thyroid tumors and have a true functional role in ATC development needs to be addressed in future studies. In the present *XPO5-CHST9* fusion, the Exportin-like protein 1 (XPO1) domain of the gene, which is responsible for transporting RNAs and proteins from the nucleus, was retained. There is currently one XPO1-inhibiting compound undergoing phase I clinical trials in patients with relapsed solid tumors [38]. Interestingly, the *CHST9* gene also harbor potential oncogenic features, as it is commonly amplified in various neoplasia [39,40]. Even so, the functional consequences of this *XPO5-CHST9* fusion are unknown.

In terms of mutational profiles, our analyses suggest the involvement of cytidine deaminases and kataegis in ATC. This is a recently described localized mutational aggregation in which multiple same-strand substitutions are clustered over kilobase-sized DNA regions [30,41,42]. As kataegis can result from AID/APOBEC-catalyzed cytidine deamination, the finding of an ATC-prevalent mutational profile suggesting dysregulation of APOBEC is highly interesting [42]. Although the potential functional and clinical consequences of kataegis in ATC are not known, the known coupling between the hypermutatibility and efficacy of immune checkpoint inhibitors makes this phenomenon highly relevant in terms of evaluating ATC patients’ responses to PDL1 inhibition [6,43,44,45]. However, it should also be mentioned that our patient cohort has undergone extensive neo-adjuvant radio- and chemotherapy prior to surgical intervention, and an iatrogenic induction of genetic abnormalities in the tumor tissue (including hypermutability) can therefore not be entirely excluded [46]. However, in a previous study, ATCs with neoadjuvant treatment did not differ in genetic composition from therapeutically naïve cases, possibly arguing against such a phenomenon [11].

WGS analyses give researchers the chance to interrogate sequences within promoter and enhancer regions. By signature analyses, we found several mutations within regulatory regions of genes associated to the Insulin-like Growth Factor-2 (IGF-2) mRNA-Binding Protein (IGF2BPs/IMPs/VICKZs) pathways, including *ACTB*, *IGF2*, and *CD44*. Interestingly, the IGF2 mRNA-binding protein 1 (IGF2BP1) was recently shown to be upregulated in the majority of ATCs, while absent in poorly differentiated and well-differentiated thyroid carcinoma [47]. Combined, these data suggest that the IGF molecular axis is of importance for ATC progression, and our findings of somatic mutations in non-coding regulatory elements might therefore warrant future interest.

We also describe two additional *TERT* promoter mutations not previously described in ATC, occurring in one sample each. We denote these alterations “G105A” (found in sample 105) and “G168A” (found in case 101), in analogy with the chosen nomenclature for the previously described “C228T” and “C250T” mutations. These variants were not annotated in conventional SNP libraries, and their somatic origin in this study could potentially suggest potential tumor-specific roles. While the conventional *TERT* promoter mutations C228T and C250T are known to recruit specific transcription factors and enhance *TERT* gene output, we do not yet know if the observed G105A and G168A variants behave in the same manner. Further studies are needed to determine the functional consequence of these *TERT* promoter mutations on *TERT* promoter activity.

Finally, given the dismal prognosis of the disease, we sought to analyze whether molecularly targetable mutations were present in our ATC cohort. We found that all ATCs interrogated displayed mutations in genes that themselves are known targets to various compounds. These results corroborate previous findings using NGS analyses to pinpoint potentially actionable genes in ATC and support the current treatment guidelines suggesting molecular analyses of ATCs in order to tailor the treatment for each patient [48,49,50].

The rather small sample size of this study is a clear limitation but must be weighed against the comprehensive bioinformatics and data yield retrieved by WGS as opposed to WES screening. Additionally, the limited number of cases prevent any significant observations regarding expressional clustering using RNA sequencing, and therefore the latter analysis was merely included to provide mRNA data for gene fusion analyses and specific copy number aberrations. Additionally, global miRNA levels could not be interrogated, and we therefore lack the ability to correlate our findings to the miRNA landscape of these tumors, which could have been fruitful given the observed *XPO5* fusion. Future functional analyses of this gene aberrancy will therefore need to be pursued.

## 5. Conclusions

Half of the ATCs sequenced displayed combined loss of cyclin-dependent kinase inhibitor genes, suggesting that studies involving specific cyclin-dependent kinase inhibitors should be pursued in patients with tumoral loss of *CDKN2A/CDKN2B*. Moreover, we highlight focal hypermutatbility within certain chromosomal regions as a recurrent theme in ATCs, suggesting that results from immune checkpoint inhibitors might not only need confirmatory testing against global hypermutability, but also assessed in cases with focal katageis. Moreover, the finding of novel *TERT* promoter mutations, a *XPO5* gene fusion, and mutations in clinically targetable genes helps demonstrate the value of comprehensive sequencing in the era of modern medicine.

## Figures and Tables

**Figure 1 cancers-13-06340-f001:**
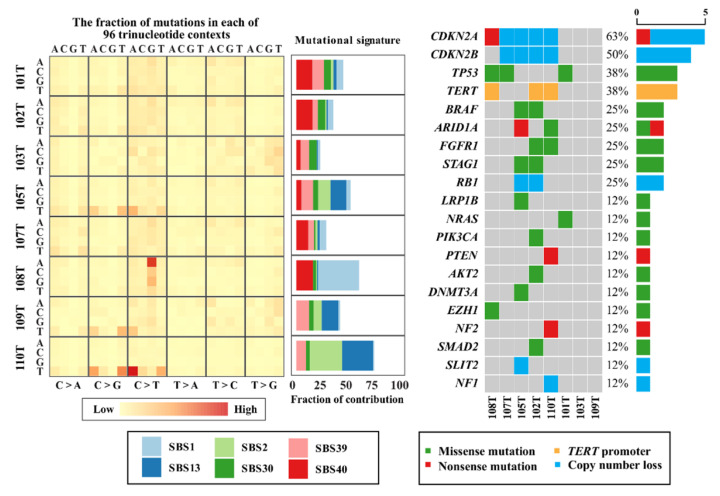
Main mutational signatures and aberrations in cancer-related genes of anaplastic thyroid carcinoma (ATC). Left: The single nucleotide variant (SNV) composition across the ATC cohort is visualized, depicting a general overrepresentation of C > T transitions. Single base substitution (SBS) signatures as defined by the Catalogue of Somatic Mutations in Cancer (COSMIC) mutational signatures are also shown, displaying an enrichment of SBS2 and SBS13 profiles characterized by the activity of Apolipoprotein B mRNA-editing enzyme, catalytic polypeptide-like (APOBEC) family of cytidine deaminases. Right: Gene aberrancy heatmap of the ATC cohort highlighting events in the top 20 mutated genes in ATC according to the COSMIC database, with frequencies in the right-most bar chart. CDKN2A was the most commonly affected gene, with four cases exhibiting copy number loss and an additional case with a nonsense mutation. Cases T103 and T109 did not harbor any events among the top 20 mutated genes in ATC according to the COSMIC database but had numerous other genetic aberrations not displayed here.

**Figure 2 cancers-13-06340-f002:**
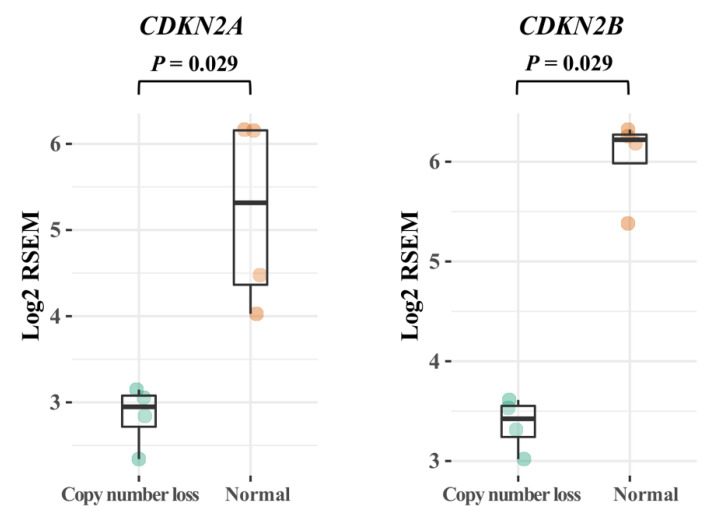
Expression of *CDKN2A* and *CDKN2B* mRNA in correlation to the corresponding gene copy number. The anaplastic thyroid carcinoma cases exhibiting synchronous copy number loss of both *CDKN2A* and *CDKN2B* (*n* = 4; 50%) display significantly lower mRNA expression of both genes compared to diploid cases (*n* = 4; 50%). *p* values < 0.05 were considered statistically significant (Mann–Whitney U).

**Figure 3 cancers-13-06340-f003:**
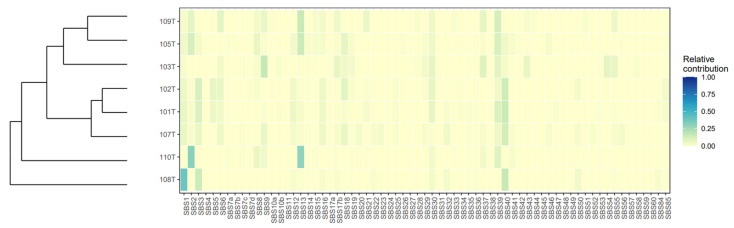
Mutational signature heatmap in anaplastic thyroid carcinoma. Single base substitution (SBS) signatures as defined by COSMIC Mutational Signatures. Tumor cases are distributed along the Y axis, with each unique mutational signature detailed along the X axis. The relative contribution of each SBS signature per case is annotated with a color scheme.

**Figure 4 cancers-13-06340-f004:**
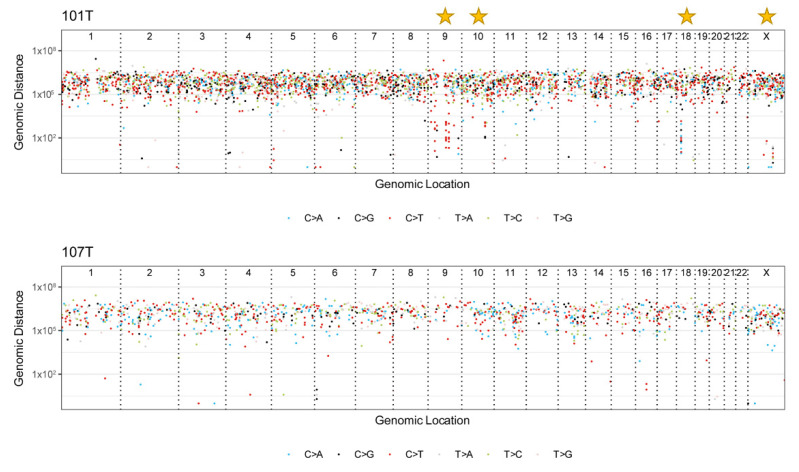
Rainfall plot visualization of kataegis-type hypermutability in anaplastic thyroid carcinoma. Cases 101 and 107 are displayed with rainfall plots, with the location of single nucleotide variant (SNV) events (chromosomal numbering) on the X axis and the genomic distance between consecutive SNV events on the Y axis. The plots help identify concentrations of SNVs as a drop in the distance between features. Case 101 displays evident areas with kataegis (focal hypermutability) as annotated by stars in this figure, while case 107 is included as a non-kataegis control. This’phenomenon was found in 4 out of 8 cases (50%).

**Figure 5 cancers-13-06340-f005:**
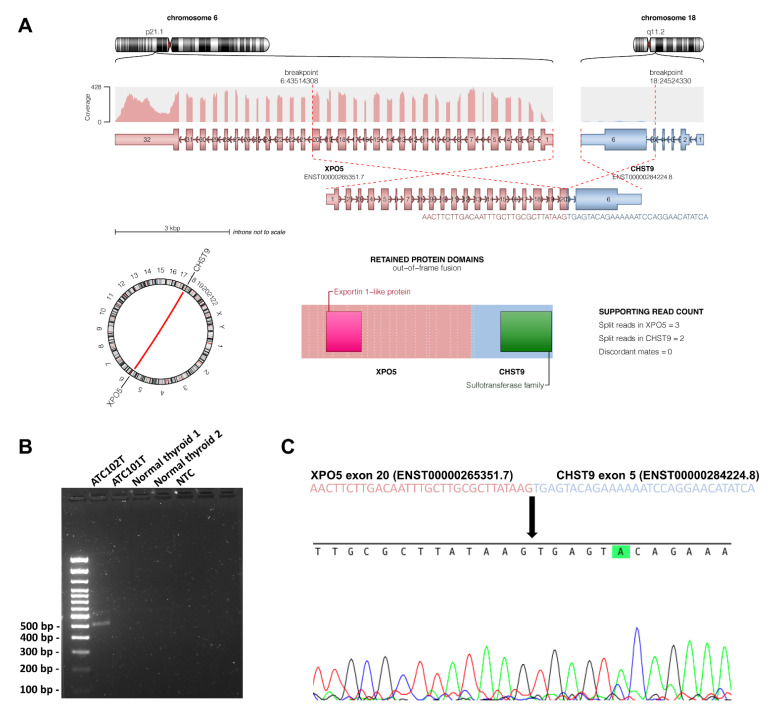
XPO5-CHST9 fusion gene validation. (**A**) Visualization of the chromosomal break points, with directions and exons annotated in the fusion gene. (**B**) Gel electrophoresis for validation of the fusion gene product in T102. T101 and two de-identified normal thyroid samples were used as negative controls. NTC; non-template control. (**C**) Sanger sequencing of the amplified fusion gene PCR product, verifying the fusion gene sequence. Arrow denotes the breakpoint.

**Table 1 cancers-13-06340-t001:** Clinical characteristics of the anaplastic thyroid carcinoma cases included in the study.

Case	Sex	Age at Surgery	Outcome	Follow-Up Time (Months)	Tumor Size (cm)	Synchronous WDTC	Tumor Cell Content (%) *
101	F	77	DOD	3	7	PTC	90
102	M	82	DOD	1	4	PTC	50
103	F	72	DOD	1	10	No	50
105	F	78	DOC	>200	8.5	No	90
107	F	89	DOD	0.5	8	PTC	90
108	F	81	DOD	4	5	No	50
109	M	70	DOD	3	11	PTC	50
110	F	92	DOD	1	6.5	No	90

WDTC: well-differentiated thyroid carcinoma; PTC: papillary thyroid carcinoma; DOD: dead of disease; DOC: dead of other cause; M: Male; F: Female; * Defined as the histopathology-proven percentage of anaplastic thyroid carcinoma cells in the tissue piece used for DNA and RNA extractions. No tissue piece sent for nucleic acid extraction exhibited a WDTC component.

## Data Availability

The datasets analyzed during the current study are available to a variable extent. Genomic datasets will not be publicly available but are available from the corresponding author upon reasonable request.

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
