# Peer review of "Pan-Genomic Sequencing Reveals Actionable CDKN2A/2B Deletions and Kataegis in Anaplastic Thyroid Carcinoma"

_cancers, 2021, doi:10.3390/cancers13246340_

Round 1

Reviewer 1 Report

In this manuscript Stenman, Yang et al. performed a pan-genomic analysis in tumor samples from a cohort of 8 patients with anaplastic thyroid carcinoma based on whole genome sequencing and RNA sequencing. The authors identified variants in established ATC-related genes as well as in 13 additional cancer genes. They also observed a focal hypermutation phenotype related to APOBEC (implied in kataegis) in 50% of the cases. The authors corroborated the importance of CDKN1A and CDKN1B in ATC. Variants were found in the promoter region of TERT, as well as a XPO5-CHST9 genes fusion. The manuscript is well written, the method is appropriated, and the conclusions are reasonable.

General comments:

This is an interesting study that bring new information in the characterization of anaplastic thyroid carcinoma.

Minor comments:

The authors should comply with the journal guidelines on genetic nomenclature, following the Human Genome Variation Society (http://varnomen.hgvs.org) and preferentially including, in the first time the mutation appear, the description at the DNA level and rs number (if available).

Author Response

Please see the attached rebuttal letter.

Reviewer 2 Report

The manuscript “Pan-genomic sequencing reveals actionable CDKN1A/2B deletions and kataegis in anaplastic thyroid carcinoma” is well organized and written.

Although the authors have performed NGS and RNA sequencing in only 8 ATC samples, which is a quite small number to represent a good cohort, they have confirmed CDKN2A/CDKN2B gene deletions, coupled with downregulation of the corresponding mRNA when compared to diploid cases. The authors have also shown regions of hypermutability and found a novel fusion gene involving XPO5 oncogene in only one case.

The article is well structured and written. Just few small notes:

- line 36: shouldn´t it be “which” instead of “with”?

- line 44: review “predominantly presents”

- line 57: review “not least”

- line 103: tissue… WAS collected

Results and particularly conclusions, could be more elaborated and explored.

Is there any explanation/alternative for why two cases (103 and 109) and no mutation/alteration whatsoever? Is this a common finding in ATCs?

Do the authors obtain any results on miRNA expression?

Are the authors addressing a potential role for the XPO5-CHST9 fusion gene on ATCs or other thyroid tumors?

As a conclusion the authors state that the majority of ATCs displayed loss of CDKNs. Although the sample size is small (n=8), it would be exactly 50% and not the majority.

Author Response

(The authors gave the same response as above.)
